# One-Class Drift Compensation for an Electronic Nose

**Xiuxiu Zhu** [1,2], **Tao Liu** [1,2,*], **Jianjun Chen** [1,2,*], **Jianhua Cao** [1,2] and **Hongjin Wang** [1,2]

1    School of Microelectronics and Communication Engineering, Chongqing University, No. 174 Shazheng Street, Shapingba District, Chongqing 400044, China; xiuxiuzhu@cqu.edu.cn (X.Z.); jianhuacao@cqu.edu.cn (J.C.); 202012021026t@cqu.edu.cn (H.W.)
2    Chongqing Key Laboratory of Bio-Perception &Intelligent Information Processing, No. 174 Shazheng Street, Shapingba District, Chongqing 400044, China
*    Correspondence: cquliutao@cqu.edu.cn (T.L.); cjj@cqu.edu.cn (J.C.)

**Abstract:** Drift compensation is an important issue in an electronic nose (E-nose) that hinders the development of E-nose's model robustness and recognition stability. The model-based drift compensation is a typical and popular countermeasure solving the drift problem. However, traditional model-based drift compensation methods have faced "label dilemma" owing to high costs of obtaining kinds of prepared drift-calibration samples. In this study, we have proposed a calibration model for classification utilizing a single category of drift correction samples for more convenient and feasible operations. We constructed a multi-task learning model to achieve a calibrated classifier considering several demands. Accordingly, an associated solution process has been presented to gain a closed-form classifier representation. Moreover, two E-nose drift datasets have been introduced for method evaluation. From the experimental results, the proposed methodology reaches the highest recognition rate in most cases. On the other hand, the proposed methodology demonstrates excellent and steady performance in a wide range of adjustable parameters. Generally, the proposed method can conduct drift compensation with limited one-class calibration samples, accessing the top accuracy among all presented reference methods. It is a new choice for E-nose to counteract drift effect under cost-sensitive conditions.

**Keywords:** electronic nose; drift compensation; domain adaptation; one-class calibration

## 1. Introduction

Over the past three decades, a bionic olfactory system named electronic nose (E-nose) has been applied to sense and identify volatilized organic compounds with a customized gas sensor array and associated intelligent algorithm models [1–4]. Behind the development of E-noses, gas sensor drift caused by inherent characteristics of metal-oxide-semiconductor sensors has played a negative role degrading the reproducibility of E-noses during long-term detections [5]. In the view of intelligent algorithm models, the gas sensor drift leads to a slowly random fluctuation of input signals, which can be seen as a data distribution movement in a multi-dimensional vector space. To maintain recognition effectiveness, drift compensation becomes an important issue, adjusting the models to adapt the time-varying data.

A number of researchers have made efforts to solve the drift problem of E-noses. In addition to straightforward attempts on gas sensor material, structure, and fabrication improvements [6–8], the algorithm approach is a popular choice counteracting the negative effect of drift. Commonly, the algorithm approach can be divided into two manners based on the usage of category information of drift correction samples. The first one is a supervised manner, using both drift correction samples and associated class information (labels) for drift compensation. The supervised manner provides complete drift information, but acquires an independent collection process to obtain sufficient and full-category drift correction samples [9–12], which leads to a costly, laborious, and time-consuming drift

compensation. To overcome this issue, researchers have tried the second manner, a flexible process allowing drift correction on fragmentary category information. Accordingly, semi-supervised learning [13,14] and active learning [15,16] methods have been introduced to use a relatively small size of full-category drift correction samples selected from massive unlabeled drift data. Once the size of labeled drift correction samples reduces to zero, that is, all drift correction samples become unlabeled data, some dimension reduction methods can be used as long as the drift disturbance is regarded as an abnormal component [17–20]. Moreover, domain adaptation has been utilized, projecting the drift data and initial training samples for a shorter distance. Following this approach, Zhang et al. enhanced the distribution consistency between drift and initial training samples in an obtained subspace to adapt drift sensor responses [21]. Yi et al. conducted a further mathematical model by using label information of the source data, distinguishing different sample classes [22]. Recently, Liu et al. have achieved an optimized data space for drift compensation with maximum label–feature correlation and minimum feature redundancy [23]. Although the second manner decreases the cost of drift compensation by removing the independent collection process of labeled drift correction samples, the movement of relative distributions between different categories has been ignored. Therefore, we need to find a new drift compensation approach overcoming the drawbacks of the above two manners.

We selected the preferred method of utilizing one-class (one-category) drift compensation instead of full-category or none drift correction samples in this study. Such one-class drift compensation not only provides definite label information to determine the relative distribution changes but also decreases the category demands of drift correction samples. Accordingly, we have established a multi-task learning model [24,25] to obtain a class-label predictor of drift data, considering the data and class label distributions of both initial training and one-class correction samples comprehensively. Specifically, domain adaptation and linear predictor model inspired us to mine the unlabeled and labeled sample information, respectively. Furthermore, we have presented an intact solution process, gaining a closed-from solution of the proposed multi-task model. We used two long-term experimental datasets from two E-nose systems as testing benchmarks. From the results on the benchmarks, the proposed method has demonstrated an obvious superiority to the other state-of-the-art methods on drift compensation.

The objectives of this study were to: (1) simplify sample preparation by using a single class of drift correction samples, (2) establish a specific mathematical model for one-class drift compensation, and (3) provide a fast solution process for the mathematical model.

The rest of this article is arranged as follows: Section 2 introduces the used drift datasets, the E-nose systems, and the details of the proposed methodology. Section 3 provides related settings, experimental results, and related discussions. Finally, the last part summarizes this study.

## 2. Materials and Methods

### 2.1. Experimental Data

In this study, we have employed two E-nose drift datasets from previous studies as the drift observations. Dataset A is a public benchmark from [26] while Dataset B is collected from an E-nose system we have designed [15].

### 2.1.1. Dataset A

This dataset was generated from an E-nose system consisting of 16 sensors of four different types (TGS2600, TGS2602, TGS2610, and TGS2620, four sensors of each type), aiming to distinguish several simple volatile organic substances in a long-term period. Eight geometric features, including two steady state features, three transient features from the rising phase, and three transient features from the declining phase, have been extracted from each gas sensor response curve. Thus, one experiment can be represented as a sample vector with 128 (16 sensors $\times$ 8 features) dimensions. In total, 13,910 samples have been collected and recorded in a 36-month long period. The testing objects include six categories,

namely, ethanol, ethylene, ammonia, acetaldehyde, acetone, and toluene. According to the time of experiments, these samples have been divided into 10 batches (shown in Table 1). The sample distributions of the 10 batches are visualized by 2-dimensional principal component analysis (PCA) plots in Figure 1.

**Table 1.** Sample information of Dataset A.

| Batch ID | Month ID | Number of Samples | | | | | | Total Number |
|---|---|---|---|---|---|---|---|---|
| | | Acetone | Acetaldehyde | Ethanol | Ethylene | Ammonia | Toluene | |
| Batch 1 | Month 1–2 | 90 | 98 | 83 | 30 | 70 | 74 | 544 |
| Batch 2 | Month 3–10 | 164 | 334 | 100 | 109 | 532 | 5 | 1244 |
| Batch 3 | Month 11–13 | 365 | 490 | 216 | 240 | 275 | 0 | 1586 |
| Batch 4 | Month 14–15 | 64 | 43 | 12 | 30 | 12 | 0 | 161 |
| Batch 5 | Month 16 | 28 | 40 | 20 | 46 | 63 | 0 | 197 |
| Batch 6 | Month 17–19 | 514 | 574 | 110 | 29 | 606 | 467 | 2300 |
| Batch 7 | Month 21 | 649 | 662 | 360 | 744 | 630 | 568 | 3613 |
| Batch 8 | Month 22–23 | 30 | 30 | 40 | 33 | 143 | 18 | 294 |
| Batch 9 | Month 24–30 | 61 | 55 | 100 | 75 | 78 | 101 | 470 |
| Batch 10 | Month 36 | 600 | 600 | 600 | 600 | 600 | 600 | 3600 |

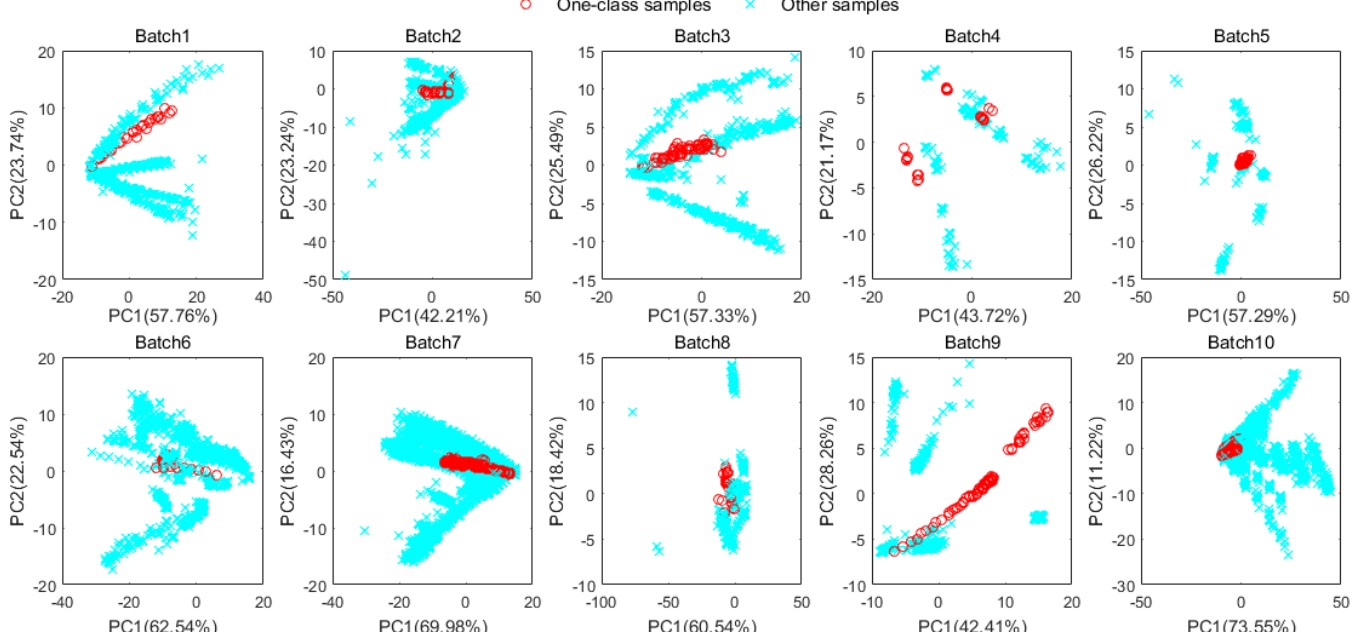

**Figure 1.** Data distributions of Batch 1–10 of Dataset A.

### 2.1.2. Dataset B

Dataset B was obtained from a self-designed E-nose system composed of 32 gas sensors [15]. The gas sensor array information is recorded in Table 2. We used this E-nose system to analyze complex aroma compounds from different beverages. Each experiment has been conducted including three phases: baseline, testing, and clean. Both baseline and testing phases lasted 3 min, maintaining the flow rate at 100 mL/min. The clean phase lasts 10 minutes with 3 L/min, the maximum flow rate of the E-nose system. Clean air was injected in both baseline and clean phases, while the headspace vapors of beverages were sampled in the testing phase. We abstracted one feature $s$ in an experiment from each sensor response curve as follows:

$$s = (R_s - R_0)/R_0 \tag{1}$$

where $R_s$ and $R_0$, respectively, denote the stable response and baseline value of a testing object. Hence, the data of one experiment were refined to a 32-dimensional sample vector considering 32 sensors in a gas sensor array. We sampled the headspace volatile compounds of seven beverages, including beer, liquor, wine, pu'erh tea, oolong tea, green tea, and black tea. With regard to each type of tea, 2 g of solid tea leaves was soaked with 200 mL of distilled water for 5 min. Afterwards, the original solution of tea can be attained by filtering out the liquid, while the original solutions of beer, liquor, and wine were bought directly from the manufacturers. Then, we formulated samples at different concentrations with both original solution and distilled water, which maintained the temperature around 25 °C. Accordingly, low, medium, and high concentration samples were formulated for each beverage according to the ratio of original solution at 14%, 25%, and 100%. Dataset B covers a 4-month experimental period, collecting 63, 189, and 189 samples in Month 1, 3, and 4, respectively. For each month, we tested seven beverages in three concentrations (14%, 25%, and 100%) created by different dilution rates. The experiments on a certain concentration were repeated one, three, and three times in Month 1, 3, and 4, respectively. Accordingly, 441 samples have been recorded in Dataset B, and we gathered these samples into Batch S1–S3 by month. Figure 2 has demonstrated the sample distributions of Batch S1–S3 in 2-dimensional PCA plots.

**Table 2.** Gas sensor information of our E-nose system.

| Model | Type | Test Objects | Model | Type | Test Objects |
|---|---|---|---|---|---|
| TGS800 | Metal oxide | Smog | MQ-7B | Metal oxide | Carbon monoxide |
| TGS813 | | Methane, ethane, propane | MQ131 | | Ozone |
| TGS816 | | Inflammable gas | MQ135 | | Ammonia, sulfide, benzene |
| TGS822 | | Ethanol | MQ136 | | Sulfuretted hydrogen |
| TGS2600 | | Hydrogen, methane | MP-3B | | Ethanol |
| TGS2602 | | Methylbenzene, ammonia | MP-4 | | Methane |
| TGS2610 | | Inflammable gas | MP-5 | | Propane |
| TGS2612 | | Methane | MP-135 | | Air pollutant |
| TGS2620 | | Ethanol | MP-901 | | Cigarettes, ethanol |
| TGS2201A | | Gasoline exhaust | WSP2110 | | Formaldehyde, benzene |
| TGS2201B | | Carbon monoxide | WSP5110 | | Freon |
| GSBT11 | | Formaldehyde, benzene | SP3-AQ2-01 | | Organic compounds |
| MQ-2 | | Ammonia, sulfide | ME2-CO | | Carbon monoxide |
| MQ-3B | | Ethanol | ME2-CH2O | Electrochemical | Formaldehyde |
| MQ-4 | | Methane | ME2-O2 | | Oxygen |
| MQ-6 | | Liquefied petroleum gas | TGS4161 | Solid electrolyte | Carbon monoxide |

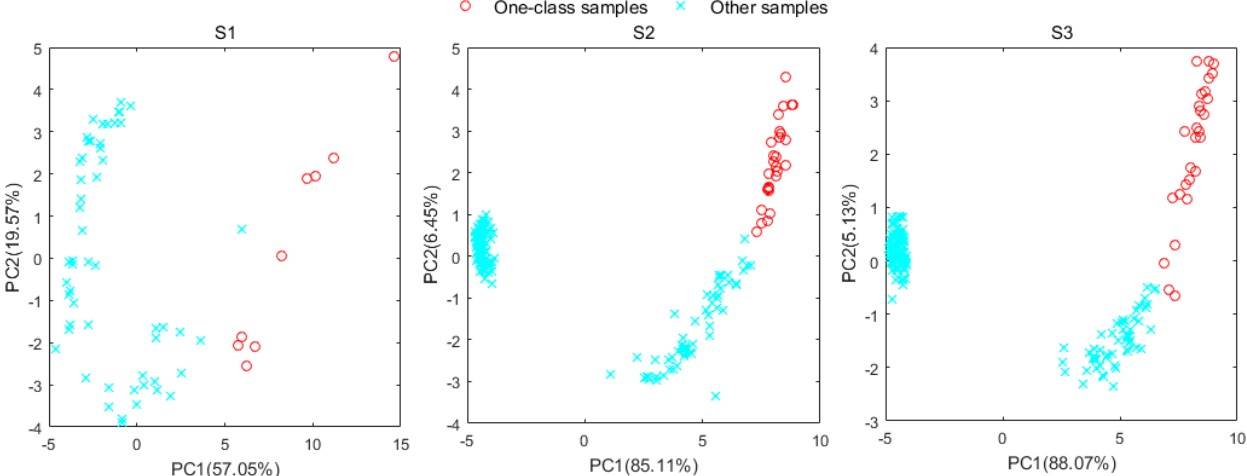

**Figure 2.** Data distribution of Batch S1–S3 of Dataset B.

### 2.2. Notations for Methods

Some specific notations should be determined for better understanding and introduction of the following models and methodologies. Primarily, the initial and following drift samples can be assumed to be two-domain data with discrepant but correlated data distribution. The domain adaptation is a kind of transfer learning paradigm, aiming to explore a common data space that makes these two-domain data be identically distributed. In this paper, we, respectively, set initial training samples $\mathbf{X}_S = \left[ x_s^1; x_s^2; \cdots; x_s^{N_S} \right] \in R^{N_S \times D}$ and drift samples $\mathbf{X}_T = \left[ x_t^1; x_t^2; \cdots; x_t^{N_T} \right] \in R^{N_T \times D}$ as the source domain and target domain data, where $D$ is the data dimension, $N_S$ and $N_T$ represent the numbers of the source domain and target domain samples. $\mathbf{Y}_S = \left[ y_s^1, y_s^2, \cdots, y_s^{N_S} \right] \in R^{N_S \times C}$ is a label matrix that contains all the class-label vectors of the source domain data, where one-hot coding (mainly uses *n*-bit status to encode *N* states. Each state is independent and only one bit is effective at any time) was used for each label vector, and *C* is the number of classes. In order to reduce the task complexity, time, and material expenditures, we tried to minimize the sizes of both calibration samples and associated sample categories. Here, we set the category size of the calibration samples to one, the minimum value we can access. We defined $\mathbf{T}_S^{(n \times D)} \subset \mathbf{X}_S$ and $\mathbf{T}_T^{(n \times D)} \subset \mathbf{X}_T$ as the calibration samples with a unique class label in the source and target domains, respectively, where *n* is a preset number of the drift correction samples (transfer samples). Moreover, we selected a first-order linear decision function to conduct classification in terms of its simple structure and low computational loads. $\mathbf{P}_S, \mathbf{P}_T \in \mathbb{R}^{D \times C}$ were two weight matrices should be solved in decision functions for the source and target domains, respectively. Additionally, we have adopted $(\cdot)^T$, $\| \cdot \|_F$, and $\| \cdot \|_*$ to represent the transpose operator, frobenius, and nuclear norms, respectively.

### 2.3. Transfer-Sample-Based Coupled Task Learning

Transfer-sample-based coupled task learning (TCTL) [27] aims to learn a prediction model for E-nose drift samples through a small number of transfer samples (drift correction samples). It is a typical cost-saving drift compensation method, and its objective function can be represented as a loss function as follows:

$$
\begin{aligned}
\min Loss(\beta_S, \beta_T) = \quad & \min_{\beta_S, \beta_T} \| \mathbf{X}_S \beta_S - y_S \|_F^2 + \lambda_1 \| \mathbf{T}_S \beta_S - \mathbf{T}_T \beta_T \|_2^2 \\
& + \lambda_2 \| \beta_S - \beta_T \|_F^2 + \lambda \sum_{j=1}^{m} w_j^2 (\beta_{S,j}^2 + \beta_{T,j}^2)
\end{aligned}
\tag{2}
$$

where $\beta_S, \beta_T \in \mathbb{R}^D$ are the source domain and target domain prediction models, respectively. $w_j$ is the deviation of $j$-th sample between the source and target domains. $\lambda$, $\lambda_1$, and $\lambda_2$ are term coefficients. In Formula (2), the first term is used to guarantee the correctness of $\beta_S$; the second and third items try to keep the similarity between two domains via recognition results and prediction models; the last one is a Tikhonov regularization term, which restores predictor information of source domain to the one of target domain. As a result, $\beta_T$ can be solved from Formula (2) as a linear predictor for drift data.

### 2.4. Transfer-Sample-Based Multiple Task Learning

In Zhang et al. [28], an improved model named transfer-sample-based multiple task learning (TMTL) was proposed by slacking the third term in TCTL's objective function. Then, the objective function of TMTL can be represented as

$$
\begin{aligned}
\min Loss(\beta_S, \beta_T) \quad &= \min_{\beta_S, \beta_T} \|\mathbf{X}_S \beta_S - y_S\|_F^2 + \tfrac{\lambda_1}{2N_T} \|\mathbf{T}_S \beta_S - \mathbf{T}_T \beta_T\|_2^2 \\
&+ \tfrac{\lambda_2}{2n} \|\mathbf{X}_S \beta_S - \mathbf{X}_S \beta_T\|_2^2 + \tfrac{\lambda}{2} \sum_{j=1}^{m} w_j^2 (\beta_{S,j}^2 + \beta_{T,j}^2)
\end{aligned}
\tag{3}
$$

where $N_T$ and $n$ are numbers of transfer samples and source domain samples, respectively. Afterwards, a standard analytical solving process can be performed, obtaining a closed-form expression of $\beta_T$ as a calibrated classifier.

### 2.5. Proposed Methodology

Both TCTL and TMTL demand multiple categories of drift correction samples, which causes extra payment of experimental materials and workloads. Therefore, we have attempted to use one-category drift correction samples for E-nose predictor updating.

#### 2.5.1. Loss Function Formulation

We aim to establish a comprehensive loss function by multi-task learning, which helps us to gain optimized $\mathbf{P}_S$ and $\mathbf{P}_T$ that projecting initial training and drift samples to a label space. Several essential demands have been considered in the modeling with one-class correction samples.

**Demand 1: empirical prediction error.** The class labels of the source domain samples can be predicted by the first-order linear model $\hat{\mathbf{Y}}_S = \mathbf{X}_S \mathbf{P}_S$, where $\hat{\mathbf{Y}}_S$ is the estimated form of $\mathbf{Y}_S$. Accordingly, we can minimize the empirical prediction error by

$$
\min_{\mathbf{P}_S} \|\hat{\mathbf{Y}}_S - \mathbf{Y}_S\|_F^2 = \min_{\mathbf{P}_S} \|\mathbf{X}_S \mathbf{P}_S - \mathbf{Y}_S\|_F^2
\tag{4}
$$

**Demand 2: rank of one-class transfer samples' labels.** The computed labels of the transfer samples in the source and target domains can be, respectively, expressed as $\hat{\mathbf{Y}}_{T_S} = \mathbf{T}_S \mathbf{P}_S$ and $\hat{\mathbf{Y}}_{T_T} = \mathbf{T}_T \mathbf{P}_T$, where $\hat{\mathbf{Y}}_{T_S}, \hat{\mathbf{Y}}_{T_T} \in R^{n \times C}$. Reasonably, both $\hat{\mathbf{Y}}_{T_S}$ and $\hat{\mathbf{Y}}_{T_T}$ should be low-rank matrices since the transfer samples are all belonging to a single class (one-hot encoding was used for classification outputs). Lower rank indicates a much purer category of the transfer samples. To maintain the class uniformity, we presented the formulation as follows:

$$
\begin{aligned}
&\min_{\mathbf{P}_S, \mathbf{P}_T} (rank(\hat{\mathbf{Y}}_{T_S}) + rank(\hat{\mathbf{Y}}_{T_T})) \\
&= \min_{\mathbf{P}_S, \mathbf{P}_T} (\|\hat{\mathbf{Y}}_{T_S}\|_* + \|\hat{\mathbf{Y}}_{T_T}\|_*) = \min_{\mathbf{P}_S, \mathbf{P}_T} (\|\mathbf{T}_S \mathbf{P}_S\|_* + \|\mathbf{T}_T \mathbf{P}_T\|_*)
\end{aligned}
\tag{5}
$$

**Demand 3: prediction error of one-class transfer samples between source and target domains**. We should guarantee the prediction correctness of the one-class transfer samples in both source and target domains via $\mathbf{P}_S$ and $\mathbf{P}_T$. In other words, ideally, the

predicted labels of the transfer samples should be equal on all two domains. To achieve this goal, we minimized the prediction error of the transfer samples as follows:

$$\min_{\mathbf{P}_S,\mathbf{P}_T}(\|\mathbf{T}_S\mathbf{P}_S - \mathbf{T}_T\mathbf{P}_T\|_F^2) \tag{6}$$

**Demand 4: dependency between samples and their class labels.** In theory, identical distributions in label space lead to similar data locations in feature space, that is, sample distributions are correlated with associate class labels. Therefore, we introduced the maximum dependency criterion (MDDM) [29] maximizing the dependency between the one-class transfer samples ($\mathbf{T}_S$ and $\mathbf{T}_T$) and their class labels ($\mathbf{L}_S$ and $\mathbf{L}_T$) by

$$
\begin{aligned}
&\max_{\mathbf{P}_S,\mathbf{P}_T} tr(\mathbf{HT}_S\mathbf{T}_S{}^T\mathbf{HL}_S) + tr(\mathbf{HT}_T\mathbf{T}_T{}^T\mathbf{HL}_T) \\
&= \max_{\mathbf{P}_S,\mathbf{P}_T} tr(\mathbf{HT}_S\mathbf{T}_S{}^T\mathbf{H}(\mathbf{T}_S\mathbf{P}_S)(\mathbf{T}_S\mathbf{P}_S)^T) + tr(\mathbf{HT}_T\mathbf{T}_T{}^T\mathbf{H}(\mathbf{T}_T\mathbf{P}_T)(\mathbf{T}_T\mathbf{P}_T)^T) \\
&= \max_{\mathbf{P}_S,\mathbf{P}_T} tr(\mathbf{HT}_S\mathbf{T}_S{}^T\mathbf{HT}_S\mathbf{P}_S\mathbf{P}_S{}^T\mathbf{T}_S{}^T) + tr(\mathbf{HT}_T\mathbf{T}_T{}^T\mathbf{HT}_T\mathbf{P}_T\mathbf{P}_T{}^T\mathbf{T}_T{}^T) \\
&= \max_{\mathbf{P}_S,\mathbf{P}_T} tr(\mathbf{P}_S{}^T\mathbf{T}_S{}^T\mathbf{HT}_S\mathbf{T}_S{}^T\mathbf{HT}_S\mathbf{P}_S + \mathbf{P}_T{}^T\mathbf{T}_T{}^T\mathbf{HT}_T\mathbf{T}_T{}^T\mathbf{HT}_T\mathbf{P}_T)
\end{aligned}
\tag{7}
$$

where $\mathbf{H} = \mathbf{I} - \frac{1}{N}\mathbf{ee}^T$, $\mathbf{e}$ is an all-one column vector.

**Demand 5: correlation between the decision functions of different domains.** The task of the decision functions $\mathbf{P}_S$ and $\mathbf{P}_T$ is to recognize discrepancy and correlation data from the source and target domains. It is bound to generate similar $\mathbf{P}_S$ and $\mathbf{P}_T$. Thus, a certain degree of similarity between $\mathbf{P}_S$ and $\mathbf{P}_T$ must be reserved as follows:

$$\min_{\mathbf{P}_S,\mathbf{P}_T}\|\mathbf{P}_S - \mathbf{P}_T\|_F^2 \tag{8}$$

**Total loss function:** combining Demand 1–5 (represented by Formulas (4)–(8)), we can obtain a loss function named one-class drift compensation model (ODCM) as follows:

$$
\begin{aligned}
minLoss(\mathbf{P}_S,\mathbf{P}_T) &= \min_{\mathbf{P}_S,\mathbf{P}_T}(\|\mathbf{X}_S\mathbf{P}_S - \mathbf{Y}_S\|_F^2 + \lambda(\|\mathbf{T}_S\mathbf{P}_S\|_* + \|\mathbf{T}_T\mathbf{P}_T\|_*) + \lambda_1\|\mathbf{T}_S\mathbf{P}_S - \mathbf{T}_T\mathbf{P}_T\|_F^2 \\
&\quad -\lambda_2 tr(\mathbf{P}_S{}^T\mathbf{T}_S{}^T\mathbf{HT}_S\mathbf{T}_S{}^T\mathbf{HT}_S\mathbf{P}_S + \mathbf{P}_T{}^T\mathbf{T}_T{}^T\mathbf{HT}_T\mathbf{T}_T{}^T\mathbf{HT}_T\mathbf{P}_T) \\
&\quad +\lambda_3\|\mathbf{P}_S - \mathbf{P}_T\|_F^2 + \lambda_4(\|\mathbf{P}_S\|_F^2 + \|\mathbf{P}_T\|_F^2))
\end{aligned}
\tag{9}
$$

where $\lambda, \lambda_1, \lambda_2, \lambda_3, \lambda_4 > 0$ are adjustable coefficients for the terms of the ODCM model, $\|\mathbf{P}_S\|_F$ and $\|\mathbf{P}_T\|_F$ are two regular terms used to prevent overfitting. Based on Formula (9), both $\mathbf{P}_S$ and $\mathbf{P}_T$ can be determined. Finally, $\mathbf{P}_T$ is the decision function to be solved, classifying drift samples $\mathbf{X}_T$ by $\mathbf{X}_T\mathbf{P}_T$.

### 2.5.2. Solution

In order to gain a closed-form solution of $\mathbf{P}_T$, we primarily converted Formula (9) to the following formation:

$$
\begin{aligned}
minLoss(\mathbf{P}_S,\mathbf{P}_T) &= \min_{\mathbf{P}_S,\mathbf{P}_T}\|\mathbf{X}_S\mathbf{P}_S - \mathbf{Y}_S\|_F^2 + \lambda(\|\mathbf{T}_S\mathbf{P}_S\|_* + \|\mathbf{T}_T\mathbf{P}_T\|_*) + \lambda_1\|\mathbf{T}_S\mathbf{P}_S - \mathbf{T}_T\mathbf{P}_T\|_F^2 \\
&\quad -\lambda_2 tr(\mathbf{P}_S{}^T\mathbf{T}_S{}^T\mathbf{HT}_S\mathbf{T}_S{}^T\mathbf{HT}_S\mathbf{P}_S + \mathbf{P}_T{}^T\mathbf{T}_T{}^T\mathbf{HT}_T\mathbf{T}_T{}^T\mathbf{HT}_T\mathbf{P}_T) + \lambda_3\|\mathbf{P}_S - \mathbf{P}_T\|_F^2 + \lambda_4(\|\mathbf{P}_S\|_F^2 + \|\mathbf{P}_T\|_F^2) \\
&= \min_{\mathbf{P}_S,\mathbf{P}_T} tr(\mathbf{X}_S\mathbf{P}_S - \mathbf{Y}_S)^T(\mathbf{X}_S\mathbf{P}_S - \mathbf{Y}_S) + \lambda tr((\mathbf{T}_S\mathbf{P}_S)^T(\mathbf{T}_S\mathbf{P}_S) + (\mathbf{T}_T\mathbf{P}_T)^T(\mathbf{T}_T\mathbf{P}_T)) \\
&\quad +\lambda_1 tr(\mathbf{T}_S\mathbf{P}_S - \mathbf{T}_T\mathbf{P}_T)^T(\mathbf{T}_S\mathbf{P}_S - \mathbf{T}_T\mathbf{P}_T) - \lambda_2 tr(\mathbf{P}_S{}^T\mathbf{T}_S{}^T\mathbf{HT}_S\mathbf{T}_S{}^T\mathbf{HT}_S\mathbf{P}_S + \mathbf{P}_T{}^T\mathbf{T}_T{}^T\mathbf{HT}_T\mathbf{T}_T{}^T\mathbf{HT}_T\mathbf{P}_T) \\
&\quad +\lambda_3 tr((\mathbf{P}_S - \mathbf{P}_T)^T(\mathbf{P}_S - \mathbf{P}_T)) + \lambda_4 tr((\mathbf{P}_S)^T(\mathbf{P}_S) + (\mathbf{P}_T)^T(\mathbf{P}_T))
\end{aligned}
\tag{10}
$$

Then, we made partial derivatives of Formula (10) with respect to $\mathbf{P}_S$ and $\mathbf{P}_T$. Letting the derivatives be 0, we can achieve:

$$
\begin{aligned}
\frac{\partial J}{\partial P_S} &= \mathbf{X}_S{}^T\mathbf{X}_S\mathbf{X}_S\mathbf{P}_S + (\lambda_1 - \lambda_2)\mathbf{T}_S\mathbf{T}_S{}^T\mathbf{P}_S + (\lambda_3 + \lambda_4)\mathbf{P}_S - \lambda\mathbf{T}_S{}^T\mathbf{H}_S\mathbf{T}_S\mathbf{T}_S{}^T\mathbf{H}_S\mathbf{T}_S\mathbf{P}_S \\
&\quad -\lambda_1\mathbf{T}_S{}^T\mathbf{T}_T\mathbf{P}_T - \lambda_3\mathbf{P}_T - \mathbf{X}_S{}^T\mathbf{Y}_S \\
&= \mathbf{A}\mathbf{P}_S + \mathbf{B}\mathbf{P}_T - \mathbf{X}_S{}^T\mathbf{Y}_S = 0
\end{aligned}
\tag{11}
$$

$$
\begin{aligned}
\frac{\partial J}{\partial \mathbf{P}_T} = \ & (\lambda_1 - \lambda_2)\mathbf{T}_T\mathbf{T}_T{}^T\mathbf{P}_T + (\lambda_3 + \lambda_4)\mathbf{P}_T - \lambda\mathbf{T}_T{}^T\mathbf{H}_T\mathbf{T}_T\mathbf{T}_T{}^T\mathbf{H}_T\mathbf{T}_T\mathbf{P}_T \\
& - \lambda_1\mathbf{T}_T{}^T\mathbf{T}_S\mathbf{P}_S - \lambda_3\mathbf{P}_S \ = \mathbf{C}\mathbf{P}_S + \mathbf{D}\mathbf{P}_T = 0
\end{aligned}
\tag{12}
$$

where

$$
\begin{aligned}
\mathbf{A} &= \mathbf{X}_S{}^T\mathbf{X}_S + (\lambda_1 - \lambda_2)\mathbf{T}_S\mathbf{T}_S{}^T + (\lambda_3 + \lambda_4)\mathbf{I} - \lambda\mathbf{T}_S{}^T\mathbf{H}_S\mathbf{T}_S\mathbf{T}_S{}^T\mathbf{H}_S\mathbf{T}_S \\
\mathbf{B} &= -\lambda_1\mathbf{T}_S{}^T\mathbf{T}_T - \lambda_3\mathbf{I} \\
\mathbf{C} &= -\lambda_1\mathbf{T}_T{}^T\mathbf{T}_S - \lambda_3\mathbf{I} \\
\mathbf{D} &= (\lambda_1 - \lambda_2)\mathbf{T}_T\mathbf{T}_T{}^T + (\lambda_3 + \lambda_4)\mathbf{I} - \lambda\mathbf{T}_T{}^T\mathbf{H}_T\mathbf{T}_T\mathbf{T}_T{}^T\mathbf{H}_T\mathbf{T}_T
\end{aligned}
\tag{13}
$$

Therefore, we can consider Formulas (12) and (13) as a pair of equations with $\mathbf{P}_S$ and $\mathbf{P}_T$. The closed-form solution of the ODCM model can be obtained by

$$
\begin{pmatrix} \mathbf{P}_S \\ \mathbf{P}_T \end{pmatrix} = \begin{pmatrix} \mathbf{A} & \mathbf{B} \\ \mathbf{C} & \mathbf{D} \end{pmatrix}^{-1} \begin{pmatrix} \mathbf{X}_S{}^T\mathbf{Y}_S \\ 0 \end{pmatrix}
\tag{14}
$$

## 3. Results and Discussions

### 3.1. Validation Settings

#### 3.1.1. Data Arrangement

We defined two settings (shown in Table 3) to restructure the drift datasets for various validation scenarios. In Table 3, *K* is the total number of batches, Setting 1 represents a short-term drift scenario with varied initial training samples and following drift samples in a relatively short period of time, while Setting 2 simulates a long-term scenario with fixed initial training samples and durative drift samples. Finally, we use "*X-Y*" to represent a certain scenario, in which *X* and *Y* are batch serial numbers corresponding to the initial training and drift samples, respectively.

**Table 3.** Scenario setting.

| Scenario | Source Domain | Target Domain |
|---|---|---|
| Setting 1 | Batch $i$ ($i = 1, 2, \ldots, K - 1$) | Batch ($i + 1$) |
| Setting 2 | Batch 1 | Batch $i$ ($i = 2, \ldots, K$) |

#### 3.1.2. Parameter Optimization

Before validation, a number of the pre-settable parameter should be optimized. For the proposed ODCM methodology, three types of parameters should be preset before usage: adjustable coefficient, number, and category of the one-class transfer samples. We used the grid search method to optimize the adjustable coefficients in the range $[10^{-4}, 10^4]$. The grid size is flexible and selected from $\{10^{-4}, 10^{-3}, 10^{-2}, 10^{-1}, 1, 10, 10^2, 10^3\}$ according to parameter scales. We chose ethylene (the fourth category) as the one-class transfer samples for Dataset A, because it appeared in all batches with relatively small quantities. For Dataset B, considering all categories are equal in quantity, we chose pu'erh tea (the fourth category) as the one-class transfer samples arbitrarily. In terms of Dataset A, Batch 6 contained 29 transfer samples (the least number among all batches), which limited the transfer sample size up to 29 in following validation. Considering that more transfer samples provide more accurate drift information, we set $n = 29$ for Dataset A. For Dataset B, we set $n = 9$ due to the fact that nine transfer samples existed in all batches.

#### 3.1.3. Reference Methods

We have employed three representative drift compensation methods as reference methods: common component PCA (CCPCA) [30], TCTL, and TMTL. All the three methods can be conducted with one-class transfer samples according to their principles, which ensures the fairness of the following evaluation. Among them, CCPCA is a classic measure to abstract signals from the drift background without any labeled drift correction samples. Considering CCPCA is a preprocessing method, we adopted a popular classification model

named support vector machine (SVM) for recognition. We used the linear kernel for the adopted SVM due to the fast speed and satisfying performance. The penalty coefficient of SVM was set to $10^{-4}$ after grid optimization. On the other hand, TCTL and TMTL are two state-of-the-art algorithm approaches based on transfer samples. Traditionally, CCPCA, TCTL, and TMTL adopt multi-class transfer samples during a drift compensation process. To adapt the one-class acquirement, we had to restrict the transfer samples of CCPCA, TCTL, and TMTL to one class with identical settings (category and quantity) to the proposed methodologies and name these methods CCPCA+, TCTL+, TMTL+. Specifically, all the algorithm parameters of one-class type methods were optimized as Section 3.1.2 illustrated. In addition, all the mentioned methodologies have been realized and implemented on Matlab 2018.

### 3.2. Recognition Results and Analysis

We assess the drift compensation performance of ODCM and other reference methods by drift sample recognition rate. Here, a higher recognition rate means a greater drift compensation effect. We have gathered all the recognition rates under different scenario settings and datasets in Tables 4–6.

**Table 4.** Recognition rate on Dataset A with short-term settings (%).

| Method | 1–2 | 2–3 | 3–4 | 4–5 | 5–6 | 6–7 | 7–8 | 8–9 | 9–10 | Average |
|---|---|---|---|---|---|---|---|---|---|---|
| CCPCA | 89.95 | 98.68 | 90.68 | 97.97 | 72.91 | 88.54 | 90.48 | 85.96 | 30.69 | 82.87 |
| TCTL | 96.95 | 99.05 | 99.34 | **99.47** | 78.78 | 78.43 | 88.38 | 97.39 | 78.42 | 90.69 |
| TMTL | 97.35 | 98.98 | 99.34 | 99.47 | 78.95 | 97.17 | 95.42 | 96.30 | 71.76 | 92.75 |
| CCPCA+ | 34.11 | 70.73 | 48.13 | 71.43 | 72.68 | 52.85 | 66.79 | 54.16 | 42.34 | 57.02 |
| TCTL+ | 98.15 | 99.05 | 100.00 | 99.34 | 76.22 | 99.44 | 96.55 | 84.30 | 71.87 | 91.66 |
| TMTL+ | **98.94** | **99.63** | 100.00 | 99.34 | **96.00** | 99.37 | 95.40 | 98.99 | 62.21 | 94.43 |
| ODCM | 98.41 | 99.48 | **100.00** | 99.34 | 78.42 | **99.72** | **96.93** | **100.00** | **90.99** | **95.92** |

**Table 5.** Recognition rate on Dataset A with long-term settings (%).

| Method | 1–2 | 1–3 | 1–4 | 1–5 | 1–6 | 1–7 | 1–8 | 1–9 | 1–10 | Average |
|---|---|---|---|---|---|---|---|---|---|---|
| CCPCA | 89.55 | 82.28 | 59.01 | 60.91 | 63.70 | 38.64 | 22.45 | 45.96 | 35.86 | 55.37 |
| TCTL | 96.95 | 96.85 | 91.30 | **98.98** | 86.78 | 82.51 | 86.05 | 83.19 | 65.75 | 87.60 |
| TMTL | 97.35 | 98.80 | 90.06 | 98.48 | 95.35 | **91.50** | **91.84** | **96.38** | 71.56 | 92.37 |
| CCPCA+ | 34.11 | 61.07 | 55.63 | 46.94 | 58.20 | 46.95 | 22.18 | 24.73 | 38.20 | 43.11 |
| TCTL+ | 98.15 | 98.22 | 98.47 | 94.70 | 69.97 | 75.18 | 78.93 | 67.34 | 80.14 | 84.57 |
| TMTL+ | **98.94** | 99.11 | 98.47 | 93.38 | 78.47 | 71.07 | 80.84 | 77.47 | 75.85 | 85.96 |
| ODCM | 98.41 | **99.26** | **100.00** | 98.68 | **97.21** | 79.16 | 84.67 | 89.37 | **91.33** | **93.12** |

**Table 6.** Recognition rate on Dataset B (%).

| Method | Setting 1 | | | Setting 2 | | |
|---|---|---|---|---|---|---|
| | S1–S2 | S2-S3 | Average | S1–S2 | S1–S3 | Average |
| CCPCA+ | 30.36 | 22.87 | 26.62 | 30.36 | 14.36 | 22.36 |
| TCTL+ | 35.19 | 59.26 | 47.23 | 35.19 | 41.98 | 38.59 |
| TMTL+ | 33.33 | **62.96** | 48.15 | 33.33 | 40.12 | 36.73 |
| ODCM | **49.38** | 61.11 | **55.25** | **49.38** | **58.64** | **54.01** |

The ODCM method achieves the highest average recognition rate in both scenario settings on Dataset A. It infers that the ODCM is stronger in robustness than all the reference methods. From Table 4, the recognition rate of ODCM reaches 90.99% in Scenario "9–10". It is 12.57% higher than the runner-up method, TCTL. As well, in Scenario "1–10", ODCM gain a recognition score 91.33%, 11.19% higher than the second one (as shown in Table 5). Upon reference methods, the results demonstrate rare discrepancy between multi-class and

one-class type methods, which is reasonable because these reference methods are designed for universal usages.

The recognition rate of each method on Dataset B is demonstrated in Table 6. Similarly, the recognition rate of the ODCM method is the favorite one under both scenario settings, 7.10% and 15.42% higher than the second-place methods at average recognition rate. It is clearly confirmed that the learned model by the proposed ODCM can reduce the negative effect of drift on recognition results.

### 3.3. Parameter Sensitivity Analysis

We intend to assess the suitability and robustness of the proposed methodology through the sensitivity analysis of the settable parameters. For the ODCM model, the adjustable coefficients $\{\lambda, \lambda_1, \lambda_2, \lambda_3, \lambda_4\}$ and the number of transferred samples $n$ are variable parameters of the model. In order to observe the performance impact of these two coefficients, we optimized them in the range: $\lambda, \lambda_1, \lambda_2, \lambda_3, \lambda_4 \in \left\{10^k, k = -4, -3, \ldots, 3, 4\right\}$, $n = \{0, 2, 4, \ldots, 20\}$ (Dataset A) and $n = \{0, 1, 2, \ldots, 9\}$ (Dataset B). If one coefficient varied, the others were fixed at the optimal value. We selected two representative scenarios, "3–4" of Dataset A and "S1–S3" of Dataset B, to observe the performance movement along with $\{\lambda, \lambda_1, \lambda_2, \lambda_3, \lambda_4\}$ and $n$. The influences on the recognition rate of the adjustable coefficients $\{\lambda, \lambda_1, \lambda_2, \lambda_3, \lambda_4\}$ are shown in Figure 3. It can be seen that the performance keeps stable in a wind range according to $\lambda$ and $\lambda_1$–$\lambda_3$. But for $\lambda_4$, the recognition accuracy fluctuates drastically, which shows that the corresponding regularization term $(\|\mathbf{P}_S\|_F + \|\mathbf{P}_T\|_F)$ plays a vital role in this model. Additionally, parameter $\lambda$ has the least impact on recognition accuracy. Figure 4 demonstrates the average recognition accuracy with the number of transfer samples $n$. The proposed ODCM methodology has the highest average accuracy in the settable range of $n$. If $n$ increases, the average accuracy is also improved. When $n$ reaches a certain degree, the recognition rate just shows a slight change. As a result, 8 and 4 were the most suitable choices for the number transfer samples considering computational cost and recognition performance for Dataset A and B, respectively.

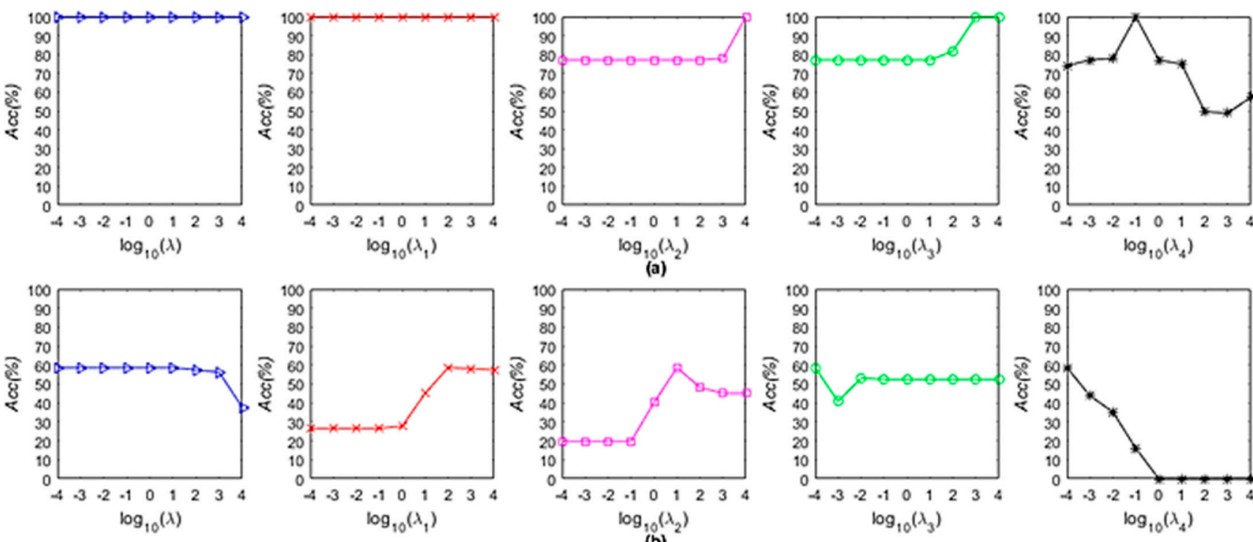

**Figure 3.** Accuracy variation with adjustable coefficients: (**a**) Scenario "3–4", Dataset A. (**b**) Scenario "S1–S3", Dataset B.

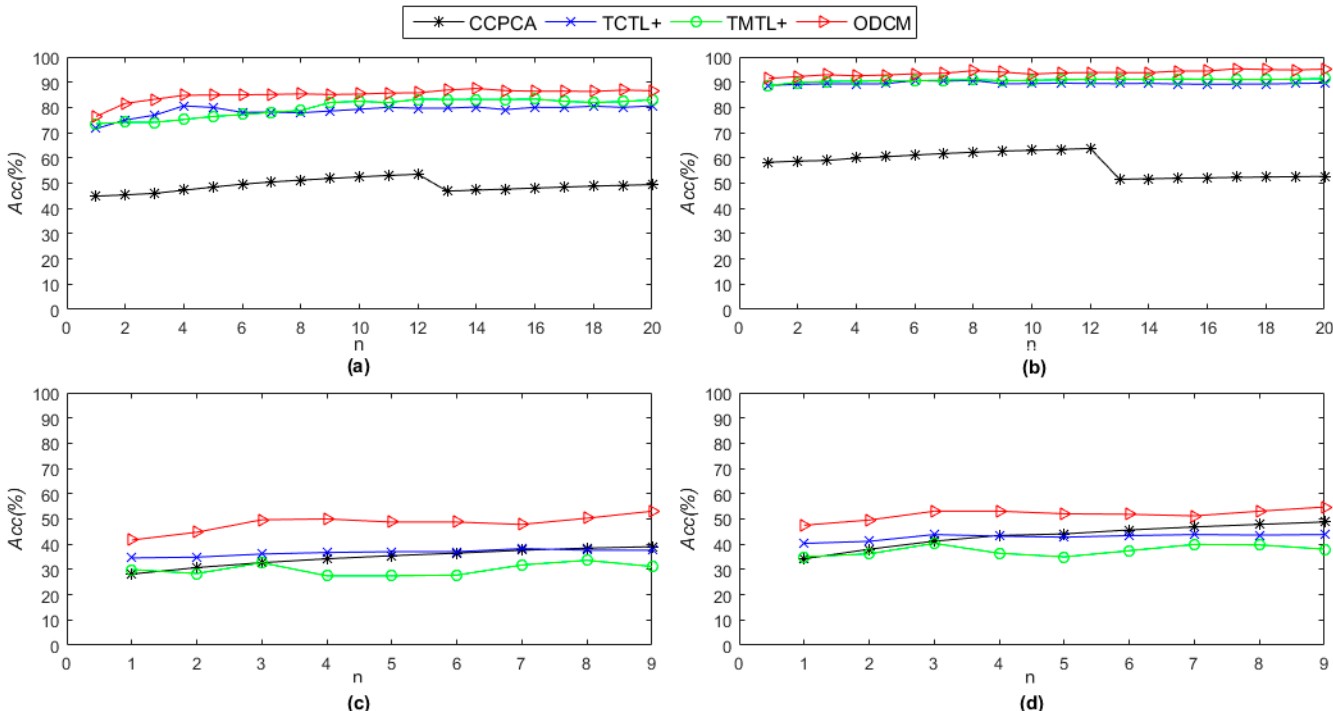

**Figure 4.** Average variation with number of transfer samples: (**a**) Setting 1, Dataset A. (**b**) Setting 2, Dataset A. (**c**) Setting 1, Dataset B. (**d**) Setting 2, Dataset B.

### 3.4. Time Complex Analysis

Implementation efficiency is an important factor that needs to be evaluated. Primarily, we have compared the theoretical time complexity between the proposed ODCM and the reference methods. For CCPCA, we should perform PCA and classifier training processes simultaneously. Therefore, we can gain the computational complexity of CCPCA as follows:

$$O_{CCPCA} = O_{PCA}\left(d^2n + d^3\right) + O_{SVM}\left(n_{sv}^3 + n \cdot n_{sv}^2 + d \cdot n \cdot n_{sv}\right) \tag{15}$$

where $d$, $n$, and $n_{sv}$ are sample dimension, quantity, and support vector quantity. According to the principle of ODCM, its computational complexity is equivalent to the ones of TCTL and TMTL. Thus, we can achieve the computational complex relation as follows:

$$O_{ODCM} = O_{TCTL} = O_{TMTL} = O\left(d^2n\right) \tag{16}$$

Based on Formulas (15) and (16), we have $O_{CCPCA} > O_{ODCM} = O_{TCTL} = O_{TMTL}$.

The computational time per sample on Dataset B have been recorded to validate above theoretical analysis in Table 7. We conducted all the methods on a computational platform with the following configuration:

CPU: Intel I5-8400
RAM: 8 GB
Hard disk: 256 GB solid-state drive
Operation system: Windows 10.

It can be seen from Table 7 that the execution time of CCPCA is much longer than other methods. This is because CCPCA requires the participation of SVM during training and testing, which is very time-consuming. However, the other three methods can give the predictor directly, so the execution time is greatly reduced. Through the average statistics, it can be known that the time complexity of TCTL, TMTL, and ODCM are at the same level, which is completely consistent with the deduction of Formulas (15) and (16).

**Table 7.** Execution time per sample of dataset B (millisecond).

| Method | S1–S2 | S1–S3 | S2–S3 | Average |
|---|---|---|---|---|
| CCPCA + SVM | 5.063 | 5.345 | 10.05 | 6.972 |
| TCTL | 0.688 | 0.732 | 0.819 | 0.746 |
| TMTL | 0.706 | 0.681 | 0.651 | 0.679 |
| ODCM | 0.704 | 0.750 | 0.814 | 0.756 |

## 4. Conclusions

In this study, a novel drift compensation manner named one-class calibration has been presented to simplify the category acquirement of drift correction sample. Based on the one-category assumption, we have proposed a specific machine learning model to learn a calibrated classifier. Moreover, we provided a closed-form solution acquisition method for the proposed model, which avoids the time-consuming iterative calculation. In addition, we used two drift datasets to validate the advantages of the proposed methodology, achieving the highest average recognition rate on one-class drift correction samples. Satisfied suitability and computational efficiency have been proven in parameter sensitivity and time complex analysis, respectively.

**Author Contributions:** Conceptualization, X.Z. and T.L.; methodology, X.Z.; software, X.Z.; validation, X.Z. and T.L.; formal analysis, X.Z. and T.L.; investigation, J.C. (Jianhua Cao) and H.W.; resources, T.L.; data curation, X.Z. writing—original draft preparation, X.Z.; writing—review and editing, T.L.; visualization, X.Z.; supervision, T.L.; project administration, J.C. (Jianjun Chen); funding acquisition, T.L. All authors have read and agreed to the published version of the manuscript.

**Funding:** This work was supported by the Open Fund of Chongqing Key Laboratory of Bioperception & Intelligent Information Processing under Grant 2019002.

**Institutional Review Board Statement:** Not applicable.

**Informed Consent Statement:** Not applicable.

**Conflicts of Interest:** The authors declare no conflict of interest.

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
