# Peer review of "One-Class Drift Compensation for an Electronic Nose"

_chemosensors, doi:10.3390/chemosensors9080208_

Round 1

Reviewer 1 Report

The authors afford the issue of drift compensation in e-noses recognition, by using a limited number of drift calibration samples belonging to a single category, imagining devices operating in remote sensing and continuous online detection. The paper is well written and data supports the conclusions about the hypothesis. However as in other papers by the authors, I suggest to include a computational complexity analysis to show the acceptable complexity of the proposed method in both theory and execution, and time of execution. 

Author Response

Q1: As in other papers by the authors, I suggest to include a computational complexity analysis to show the acceptable complexity of the proposed method in both theory and execution, and time of execution.

Answer: we have supplemented a new section “3.4 Time complex analysis” in Line 326-352 to provide the comparison of theoretical complex and execution time.

Thank you for your helpful and professional suggestions!

Reviewer 2 Report

The content of this paper is thorough in reporting all of the details for each section, but there is significant structural and writing deficiencies that need to be corrected as follows.

Major paper structural corrections needed (for reporting according to convention of a scientific paper required for this journal and most other professional journals):

Section 2 topic should be 2. Materials and Methods (not Proposed Methods)

Section 3 topic should be 3. Results (not Experiments)

Please add a new section 4. Discussion to compare your new methods and results to those reported in the literature by other scientists. All of your references in this paper are either Introduction (25 references) or Materials and Methods references (5 references). You need to compare your results to those of other scientists and provide references of key similarities and differences in results. Thus, section 5 becomes 5. Conclusions.

The first sentence of each new paragraph should be a direct topic statement that explains the main theme of the paragraph. Therefore, the structure of the first sentence should not be an indirect statement that uses either dependent clauses, adverbs, or prepositions (followed by a comma) before providing the subject of the sentence. To make a direct statement, put the subject first in the sentence followed by the dependent clause or prepositions. This is essential for a topic sentence.

The authors sometimes use words like Nowadays, (as in the first sentence of the Introduction) which is a form of conversational language that has no place in scientific writing. Conventional scientific writing should be written in a formal professional way with adequate grammar and not using conversational language or excessive indirect statements (especially for topic sentences of a new paragraph). Please correct this problem throughout the entire manuscript.

Results

Some of the Figures are too small to be easily read or interpreted and thus ineffective (such as Figures 1 and 2).

Figure 3 uses italics for the x- and y-axis to graphs. Italics is very hard to read and this should be corrected to normal type that is easier to read.

References

Reference 3 is not cited correctly. It should be Wilson, A.D. (for the author)

Author Response

Q1: Major paper structural corrections needed (for reporting according to convention of a scientific paper required for this journal and most other professional journals):

Section 2 topic should be 2. Materials and Methods (not Proposed Methods)

Section 3 topic should be 3. Results (not Experiments)

Please add a new section 4. Discussion to compare your new methods and results to those reported in the literature by other scientists. All of your references in this paper are either Introduction (25 references) or Materials and Methods references (5 references). You need to compare your results to those of other scientists and provide references of key similarities and differences in results. Thus, section 5 becomes 5. Conclusions.

Answer: we have revised the whole structure of this paper. Considering the results and discussions are closely integrated in this paper, we finally divided this paper into four sections:

Section 1 Introduction

Section 2 Materials and methods

Section 3 Results and discussions

Section 4 Conclusions

In Section 2, we arranged contents including experimental data (E-nose drift data), used reference methods and the proposed methodology.

In Section 3, assessment settings, recognition comparison, parameter sensitivity and time complex analysis have been gathered together.

Q2: The first sentence of each new paragraph should be a direct topic statement that explains the main theme of the paragraph. Therefore, the structure of the first sentence should not be an indirect statement that uses either dependent clauses, adverbs, or prepositions (followed by a comma) before providing the subject of the sentence. To make a direct statement, put the subject first in the sentence followed by the dependent clause or prepositions. This is essential for a topic sentence.

Answer: we have carefully checked all the first sentence of paragraphs and revised the inappropriate ones. The revised ones are labeled with red color.

Q3: The authors sometimes use words like Nowadays, (as in the first sentence of the Introduction) which is a form of conversational language that has no place in scientific writing. Conventional scientific writing should be written in a formal professional way with adequate grammar and not using conversational language or excessive indirect statements (especially for topic sentences of a new paragraph). Please correct this problem throughout the entire manuscript.

Answer: we have changed “Nowadays” to “Over the past three decades”. Besides, we also have tried our best to find and modify such improper uses of words.

Q4: Some of the Figures are too small to be easily read or interpreted and thus ineffective (such as Figures 1 and 2).

Figure 3 uses italics for the x- and y-axis to graphs. Italics is very hard to read and this should be corrected to normal type that is easier to read.

Answer: we have redrawn Figure 1-3 according to your suggestions.

Q5: Reference 3 is not cited correctly. It should be Wilson, A.D. (for the author).

Answer: we have revised it.

Thank you for your helpful and professional suggestions!

Reviewer 3 Report

The paper under review describes a protocol for drift compensation in gas sensor arrays. While the topic is interesting and important, the quality of presentation can be significantly improved in my opinion. I have the following concerns regarding the manuscript:

1) Throughout the manuscript the authors employ unconventional terminology. For example, "drift calibration" is an improper term. Please use drift correction, or drift compensation instead.

2) The introduction is very sketchy and does not explain the details, the drawbacks and the benefits of other proposed drift correction methods. The authors are focused on a strange “with label” and “without label” situations, while the task of drift compensation is more general and has the same nature for both classification and quantitative analysis problems. In case of unsupervised learning (“without label”) the purpose of drift compensation is not clear, as this type of models is normally employed for exploratory purposes only where no long-term measurements are typically assumed.

3) In case of dataset A the composition of the samples and the purpose of the sample analysis are not clear. Was it a plain distinguishing between different vapors? Detailed explanations are required.

4) In case of datset B the practical utility of such tests is not clear. I see no real world task that would be associated with the necessity to distinguish between beer, wine and tea. This can be done with a naked eye and application of e-nose is not justified. Detailed explanations are required. The same holds for the employed sensor array – please provide explicit description of the experiment.

5) The motivation for choosing these particular numbers of transfer samples is not clear. Normally, the results of drift correction depend a lot on a size of transfer set. Detailed explanations are required.

6) While CCPCA is indeed a widely explored method for drift correction in e-nose, the two other methods are not that popular. A detailed explanation of the math behind these methods would be very helpful for the readers.

7) In conclusion section the nature of the declared reduction of calibration cost is not clear. As far as I understand, the proposed ODCM methodology requires the same number of transfer samples as the benchmark techniques.

8) Based on the results reported in the Table 4 I am not convinced that ODCM is good for long-term drift compensation as it was outperformed by other methods. Please reconsider the conclusion section as some of the claims are not supported by the results.

9) Some other minor points are provided as sticky notes and coloring in the attached pdf.

I am advising against the acceptance of the paper in its’ present form. It can be reconsidered after a major revision.

Author Response

Response to Reviewer 3’s comments

Q1: Throughout the manuscript the authors employ unconventional terminology. For example, "drift calibration" is an improper term. Please use drift correction, or drift compensation instead.

Answer: we have changed “drift calibration” into “drift compensation” or “drift correction” in the whole manuscript.

Q2: The introduction is very sketchy and does not explain the details, the drawbacks and the benefits of other proposed drift correction methods. The authors are focused on a strange “with label” and “without label” situations, while the task of drift compensation is more general and has the same nature for both classification and quantitative analysis problems. In case of unsupervised learning (“without label”) the purpose of drift compensation is not clear, as this type of models is normally employed for exploratory purposes only where no long-term measurements are typically assumed.

Answer: we have rewritten the corresponding part, changed the terms “with label” and “without label” to “supervised manner” and “flexible process”, and explained the principle, advantages and disadvantages of the “supervised manner” in Line 40-46 as follows:

“Commonly, the algorithm approach can be divided into two manners based on the usage of category information of drift correction samples. The first one is a supervised manner, using both drift correction samples and associated class information (labels) for drift compensation. The supervised manner provides complete drift information, but acquires an independent collection process to obtain sufficient and full-category drift correction samples [9–12], which leads to a costly, laborious and time-consuming drift compensation.”

On the other hand, to clearly illustrate the principle, advantages and disadvantages of the “flexible manner”, we have added related contents in Line 46-51 and 60-62 as follows:

“To overcome this issue, researchers have tried the second manner, a flexible process allowing drift correction on fragmentary category information. Accordingly, semi-supervised learning [13,14] and active learning [15-16] methods have been introduced to use relatively small size of full-category drift correction samples selected from massive unlabeled drift data. Once the size of labeled drift correction samples reduces to zero, that is, all drift correction samples become unlabeled data, ……”

“Although the second manner decreases the cost of drift compensation by removing the independent collection process of labeled drift correction samples, the movement of relative distributions between different categories have been ignored.”

Q3: In case of dataset A the composition of the samples and the purpose of the sample analysis are not clear. Was it a plain distinguishing between different vapors? Detailed explanations are required.

Answer: we have carefully reviewed the quoted paper, and found that the motivation of this data was to test the drift compensation method proposed in that paper. Specifically, the negative effect of drift was assumed to be eliminated by drift compensation if the mathematical model can identify different vapors correctly. So, we have added an illustration in Line 89-90 as follows:

“……, aiming to distinguish several simple volatile organic substances in a long-term period.”

Q4: In case of dataset B the practical utility of such tests is not clear. I see no real world task that would be associated with the necessity to distinguish between beer, wine and tea. This can be done with a naked eye and application of e-nose is not justified. Detailed explanations are required. The same holds for the employed sensor array – please provide explicit description of the experiment.

Answer: we collected dataset B for the research of E-nose based beverage identification. We intended to know whether our E-noses can recognize different beverages on gas sensors, which is a first step to replace human judgment. To illustrate the above explanation in the paper, we have added related content in Line 105-111 as follows:

“We used this E-nose system to analyze complex aroma compounds from different beverages. Each experiment has been conducted including three phases: baseline, testing and clean. Both baseline and testing phases lasted 3 min., maintaining the flow rate at 100ml/min. The clean phase was a 10-min. A long process with 3L/min, the maximum flow rate of the E-nose system. Clean air was injected in both baseline and clean phases while the headspace vapors of beverages were sampled in the testing phase.”

Furthermore, the detailed experimental process is also added in Line 120-127 as follows:

“With regard to each type of tea, 2 g of solid tea leaves were soaked with 200 ml of distilled water for 5 minutes. Afterwards, the original solution of tea can be attained by filtering out the liquid, while the original solutions of beer, liquor and wine were bought directly from the manufacturers. Then, we formulated samples at different concentrations with both original solution and distilled water which maintain the temperature around 25C.Accordingly, low, medium and high concentration samples were formulated for each beverage according to the ratio of original solution at 14%, 25% and 100%.”

In addition, Table 2 has been added for the explicit description of the employed gas sensor array as follows:

Table 2. Gas sensor information of our E-nose system

Model

Type

Test objects

Model

Type

Test objects

TGS800

Metal oxide

Smog

MQ-7B

Metal oxide

Carbon monoxide

TGS813

Methane, ethane, propane

MQ131

Ozone

TGS816

 Inflammable gas

MQ135

Ammonia, sulfide, benzene

TGS822

Ethanol

MQ136

Sulfuretted hydrogen

TGS2600

Hydrogen, methane

MP-3B

Ethanol

TGS2602

Methylbenzene, ammonia

MP-4

Methane

TGS2610

Inflammable gas

MP-5

Propane

TGS2612

Methane

MP-135

Air pollutant

TGS2620

Ethanol

MP-901

Cigarettes, ethanol

TGS2201A

Gasoline exhaust

WSP2110

Formaldehyde, benzene

TGS2201B

Carbon monoxide

WSP5110

Freon

GSBT11

Formaldehyde, benzene

SP3-AQ2-01

Organic compounds

MQ-2

Ammonia, sulfide

ME2-CO

Electrochemical

Carbon monoxide

MQ-3B

Ethanol

ME2-CH2O

Formaldehyde

MQ-4

Methane

ME2-O2

Oxygen

MQ-6

Liquefied petroleum gas

TGS4161

Solid electrolyte

Carbon monoxide

Q5: The motivation for choosing these particular numbers of transfer samples is not clear. Normally, the results of drift correction depend a lot on a size of transfer set. Detailed explanations are required.

Answer: we have revised the explanation on the setting of transfer sample size in Line 258-262 as follows:

“In terms of Dataset A, Batch 6 contains 29 transfer samples (the least number among all batches), which limits the transfer sample size up to 29 in following validation. Considering that more transfer samples provide more accurate drift information, we set n=29 for Dataset A. For Dataset B, we set n=9 due to the fact that the 9 transfer samples are existed in all batches.”

Q6: While CCPCA is indeed a widely explored method for drift correction in e-nose, the two other methods are not that popular. A detailed explanation of the math behind these methods would be very helpful for the readers.

Answer: we have added Section 2.3 and 2.4 (Line 158-178) for TCTL and TMTL explanations, respectively. For CCPCA, we have added more setting details in Line 267-270 as follows:

“Considering CCPCA is a preprocessing method, we adopted a popular classification model named support vector machine (SVM) for recognition. We used the linear kernel for the adopted SVM due to the fast speed and satisfying performance. The penalty coefficient of SVM was set to 10-4 after grid optimization.”

Q7: In conclusion section the nature of the declared reduction of calibration cost is not clear. As far as I understand, the proposed ODCM methodology requires the same number of transfer samples as the benchmark techniques.

Answer: upon your comment, we have realized that the proposed method does not indeed reduce the number of transfer samples, but simplifies the sample preparation work by using a single class of transfer samples. It makes drift correction easier to carry. To illustrate this idea, we have revised the text in three places:

   Line 14-15, “……utilizing a single category of drift correction samples for more convenient and feasible operations.”

Line 65-68, “we prefer to adopt few (one category), instead of full-category or none, drift correction samples to drift compensation. Such one-class drift compensation not only provides definite label information to determine the relative distribution changes but also decreases the category demands of drift correction samples.”

   Line 355, “……to simplify the category acquirement of drift correction sample.”

Q8: Based on the results reported in the Table 4 I am not convinced that ODCM is good for long-term drift compensation as it was outperformed by other methods. Please reconsider the conclusion section as some of the claims are not supported by the results.

Answer: Yes, Table 4 does not show the ODCM outperforms to reference methods in all cases. ODCM has won in 4/9 cases and finally reached the highest average recognition rate, which indicates the ODCM has more relative advantages than the other methods. To clearly illustrate this situation, we have revised the expression of the Conclusion part in Line 359-362 as follows:

“In addition, we used two drift datasets to validate the advantages of the proposed methodology, achieving the highest average recognition rate on one-class drift correction samples. Satisfied suitability and computational efficiency have been proved in parameter sensitivity and time complex analysis, respectively.”

Q9: Some other minor points are provided as sticky notes and coloring in the attached pdf.

Answer: based on the reviewer’s comments in the attached pdf file, we have made corrections as follows:

  • Correct all the improper usage of link marks in the paper.
  • Line 32-34, change the unclear statement “as spatial distribution movement of gaseous samples” to “the gas sensor drift leads to a slowly random fluctuation of input signals, which can be seen as a data distribution movement in a multi-dimensional vector space.”
  • Line 136-137, change “initial training and drift samples” to “initial and following drift samples” for more clear expression.
  • Line 144-146, add an explanation for one-hot coding as “……mainly uses n-bit status to encode N Each state is independent and only one bit is effective at any time.”
  • Line 120-127, add detailed operation for three different concentrations.
  • Line 114, add Table 2 to show the gas sensor array information.
  • Line 277-278, add instruction of used software package as “In addition, all the mentioned methodologies have been realized and implemented on Matlab 2018”
  • Line 357-362, revise the unclear statement “Moreover, we solved the assumed classifier from the machine learning model in a closed-from manner, which makes the model be possible for real-time calibration.” to “Moreover, we provided a closed-form solution acquisition method for the proposed model, which avoids the time-consuming iterative calculation.”

Thank you for your helpful and professional suggestions!

Round 2

Reviewer 2 Report

This manuscript is significantly improved with larger figures and improvements in overall sectional structure and sentence structure. The paper could still be improved with the following suggested revisions to cover the remaining deficiencies as follows.

Additional suggested grammar and formatting revisions:

Introduction

The 3rd paragraph (p. 2, Line 65) still has an indirect statement in the first sentence of this paragraph. Change to a direct statement. We selected the preferred method of utilizing one-class (one-category) drift compensation instead of full-category or none drift correction samples in this study. 

Please provide a new paragraph (at the end of the Introduction) listing the main objectives of this study in sentence format [using, for example, the general sentence structure: The objectives of this study were to: 1)....2).... and 3)....]

Materials and Methods

p. 5  2.4, Line 171 the word ref. should be replaced with Zhang et al. [28] 

Results and Discussion

p. 8  3.2, Line 278 the words In this subsection are not needed. Change to: We assessed the drift compensation....

Discussion references (expand)

The authors have done very little if any to discuss and compare their results to those of other published research studies related to this specific subject area. Please add current references in the Discussions subsection to compare your results to other studies and to give indications of the overall significance of your work compared to what has been done previously. 

Author Response

Response to Reviewer 2’s comments

Q1: The 3rd paragraph (p. 2, Line 65) still has an indirect statement in the first sentence of this paragraph. Change to a direct statement. We selected the preferred method of utilizing one-class (one-category) drift compensation instead of full-category or none drift correction samples in this study. 

Answer: we have revised the mentioned sentence as reviewer’s suggestion in Line 65-66.

Q2: Please provide a new paragraph (at the end of the Introduction) listing the main objectives of this study in sentence format [using, for example, the general sentence structure: The objectives of this study were to: 1)....2).... and 3)....]

Answer: we have added a paragraph to list the main objectives of this study as reviewer’s suggestion in Line 78-80 as follows:

“The objectives of this study were to: 1) simplify sample preparation by using a single class of drift correction samples, 2) establish a specific mathematical model for one-class drift compensation, and 3) provide a fast solution process for the mathematical model.”

Q3: p. 5  2.4, Line 171 the word ref. should be replaced with Zhang et al. [28].

Answer: we have revised it in Line 175 as reviewer’s suggestion.

Q4: p. 8  3.2, Line 278 the words In this subsection are not needed. Change to: We assessed the drift compensation.....

Answer: we have revised it in Line 285-286 as reviewer’s suggestion.

Q5: The authors have done very little if any to discuss and compare their results to those of other published research studies related to this specific subject area. Please add current references in the Discussions subsection to compare your results to other studies and to give indications of the overall significance of your work compared to what has been done previously.

Answer: In fact, this question is also the problem we have to face in this study. There are indeed plenty of drift compensation methods in recent published studies, but, as far as we know, most of these methods cannot be implemented with a single class of transfer samples. So, we have to use only three methods as reference methods for our proposed method. To explain this reason, we have added associate content in Line 268-270 as follows:

“All the three methods can be conducted with one-class transfer samples according to their principles, which ensures the fairness of the following evaluation.”

Thank you for your helpful and professional suggestions!

Reviewer 3 Report

The authors did a good job in addressing the comments raised in the previous round of review. In my opinion the paper can be accepted for publication now.

Author Response

Thank you for your helpful and professional suggestions in previous-round review.

Round 3

Reviewer 2 Report

The manuscript is much improved with the addition of the study objectives, wording adjustments to create direct statements, and addressing and explaining the issue of a lack of available references in this specific area of data analysis which is different from those previous methods published and thus not directly comparable. Kudos to the authors who have made a good effort to correct deficiencies in the manuscript and paid attention to detail in providing these revisions.